# Comparing intramuscular adipose tissue on T1-weighted and two-point Dixon images

**Madoka Ogawa** [1,2,3¤] *, **Akito Yoshiko**[4], **Noriko Tanaka**[1,5], **Teruhiko Koike**[5,6], **Yoshiharu Oshida**[5,6], **Hiroshi Akima**[1,5]

**1** Graduate School of Education and Human Development, Nagoya University, Nagoya, Aichi, Japan,
**2** Japan Society for the Promotion of Science, Chiyoda-ku, Tokyo, Japan, **3** Nippon Sport Science
University, Setagaya-ku, Tokyo, Japan, **4** School of International Liberal Studies, Chukyo University, Nagoya,
Japan, **5** Research Center of Health, Physical Fitness and Sports, Nagoya University, Nagoya, Japan,
**6** Graduate School of Medicine, Nagoya University 65 Tsurumai, Nagoya, Aichi, Japan

¤ Current address: Nippon Sport Science University, Setagaya-ku, Tokyo, Japan
* d0c0dem0.mad020@gmail.com

**Data Availability Statement:** Our data contain potentially identifying information and ethics committee does not allow us to upload the data publicly. Data are available from Ethics Committee

## Abstract

### Purpose

The purpose of this study was to compare intramuscular adipose tissue content determined by two-point Dixon imaging and T1-weighted imaging, calculated using thresholding techniques.

### Methods

In total, 19 nonobese younger adults (26.2 ± 4.9 years) and 13 older adults (72.2 ± 6.0 years) were recruited. Axial images of the mid-thigh were taken using T1-weighted and two-point Dixon sequences with a 3.0 T whole-body magnetic resonance device and used to measure intramuscular adipose tissue content of the vastus lateralis, adductor magnus, and long head of the biceps femoris.

### Results

There was no significant difference in intramuscular adipose tissue content between T1-weighted and two-point Dixon imaging for the vastus lateralis (11.0 ± 4.4% and 12.2 ± 2.4%); however, intramuscular adipose tissue content determined by T1-weighted imaging was significantly higher than that of two-point Dixon imaging for the other muscles. Bland-Altman analysis showed a proportional bias for intramuscular adipose tissue calculations in all three muscles.

### Conclusion

The validity of intramuscular adipose tissue content measurements between T1-weighted and two-point Dixon imaging is muscle-specific. This study showed discrepancies of intra-muscular adipose tissue content between T1-weighted and two-point Dixon imaging.

for researchers who meet the criteria for access to confidential data. Ethics committee of Nagoya University Graduate School of Medicine. Address: Nagoya University 65 Tsurumai, Showa-ku, Nagoya, Aichi 466-8550, Japan. E-mail: ethics@med.nagoya-u.ac.jp

**Funding:** This study has received funding by Grant-in-Aid for JSPS Research Fellow and Suzuken Memorial Foundation and Editage Grant.

**Competing interests:** The authors have declared that no competing interests exist.

## Advances in knowledge

This study's results suggest that care should be taken when selecting an imaging modality for intramuscular adipose tissue, especially for patients who would be suspected to have higher intramuscular adipose tissue values.

## Introduction

Adipose tissue is a specialized, loose connective tissue laden with adipocytes and sometimes stored in ectopic locations, such as near muscle, liver, the heart, and within the abdominal cavity [1]. Adipose tissue usually infiltrates within or around the muscle fibers, with adipose tissue infiltration "within" the muscle referred to as intramuscular adipose tissue or "IntraMAT" [1]. IntraMAT is an important measure in research, and in older individuals and patients with Duchenne muscular dystrophy, IntraMAT is inversely associated with insulin resistance [2–5], force generation [6–9], and a range of functional ability tests (e.g., 6-minute walk distance [8, 10], gait speed [11, 12], sit-to-stand [11, 12], stair descent time [8, 12], and timed up and go tests [12]). Furthermore, studies show that IntraMAT values of the lower limb muscles, such as the thigh muscles, increase with age [6–9]. IntraMAT values therefore aid in our understanding of insulin resistance, muscle functional ability, and muscle quality and could be used to create thresholds that identify patients who may be at risk of developing type 2 diabetes in the future. The values could also help in identifying patients with early myopathy as a window into muscular dystrophy, for example.

Magnetic resonance imaging (MRI) is a powerful tool to assess the quality and quantity of human skeletal muscles, and to investigate the IntraMAT of target muscles, T1-weighted (T1W) imaging and the Dixon method have been frequently used in previous studies. IntraMAT has been calculated based on T1W images combined with segmentation analysis in previous studies [12–16]. The Dixon method is an alternate, relatively inventive technique for the evaluation of fat fractions based on the different precessional frequencies of water and lipid protons [17–19]. Previous studies confirmed that IntraMAT content by both the T1W and Dixon methods were consistent or correlated with muscle biopsy, which is the gold standard to measure IntraMAT [20, 21]. The IntraMAT content in the thigh measured by the three-point Dixon (3PD) method was approximately 3% in younger men and 6% in older men [22]. Others have reported specific muscle values: 3.0% for younger and 5.4% for older men in the vastus lateralis (VL) and 3.9% for younger and 7.2% for older men in the biceps femoris-long head (BF-L) [22]. Recently, a study reported no significant difference ($P = 0.83$) between the fat fractions determined by two-point Dixon (2PD) and lipocytes assessed by histological analysis taken from muscle biopsies, indicating that the Dixon technique is reliable for the quantification of IntraMAT [20]. However, T1W imaging has been widely used for the diagnosis of diseases as well as in medical imaging research. Previous studies with T1W imaging have used histograms to separate skeletal muscle tissue and adipose tissue based on differences in pixel intensity, which mainly comes from differences in the spin-lattice relaxation times between skeletal muscle and adipose tissues, whereas others have used different muscle and adipose tissue threshold settings for discrimination [14, 15, 23–25]. We previously reported that IntraMAT content using T1W images is 4.6% in the VL and 16.2% in the BF-L in older subjects (70.7 ± 3.8 years) [14]. These values are slightly higher than those taken by 3PD images (0.8% for VL and 9% for BF-L) [20]. Several factors affect IntraMAT content between two imaging techniques, including subject characteristics, differences in spatial resolution, and differences in the sensitivity of the imaging apparatus to adipose tissues located inside and outside of the muscle cells.

The proportion of IntraMAT by 2PD is consistent with the results of histological evaluations from biopsy specimens [20], and a correlation between the proportion of IntraMAT by T1W and biopsy results has been reported [21]. It is therefore plausible that IntraMAT values of T1W and 2PD may match; yet, to the best of our knowledge, no study has tested the agreement between T1W imaging and Dixon imaging for IntraMAT content on the same subjects using the same locations. Thus, the purpose of this study was to compare IntraMAT content determined by 2PD imaging and T1W imaging, calculated using thresholding techniques. This study is useful for judging whether there is inconsistency in the IntraMAT contents by the two methods reported in previous studies. Those results are instrumental for selecting the imaging method used for diagnosis of IntraMAT in diseased patients. We hypothesized that IntraMAT values obtained with T1W would be slightly higher than those taken using 2PD.

## Methods

### Subjects

Nineteen younger men (26.2 ± 4.9 years, 22.2 ± 2.2 kg/m$^2$) and 13 older men (72.2 ± 6.0 years, 23.9 ± 1.6 kg/m$^2$) volunteered to participate in this study. Subjects were widely recruited healthy and non-obese (BMI < 25.0 kg/m$^2$) volunteers using a poster in the library. This experiment was conducted as a prospective study in 2016. Before the experiment, the procedure, purposes, risks, use data for research and benefits associated with the study were explained and written consent was obtained with all subjects. This study was approved by the Research Ethics Committee (Nagoya University; 2016–0254) and the investigation was performed according to the principles outlined in the Declaration of Helsinki.

### MRI acquisition

Subjects were assessed with a 3.0 T whole-body MRI scanner (MAGNETOM Verio, Siemens Healthcare Diagnostics K.K., Tokyo, Japan). Subjects were placed in a supine position and images of the thigh were acquired using a body coil. We took consecutive images of the entire thigh using T1W and 2PD imaging. We defined the mid-thigh according to markers attached at the middle point between the greater trochanter and the lateral condyle of the femur. The durations for image acquisition were 4 min for the T1W and 8 min for the 2PD. T1W spin-echo transaxial images of the right thigh were collected with the following sequences: three-dimensional, TR = 604 ms; TE = 12 ms; flip angle = 120˚; optimized field of view = 256 × 256 mm; slice thickness = 10 mm; and interslice gap = 0 mm. All subjects were instructed to remain as still as possible. In-phase and out-of-phase water and fat transaxial images were obtained to create water and fat images for analysis. 2PD images of the right thigh were acquired with the following sequences: three-dimensional, TR = 20 ms; TE1 = 2.450 ms; TE2 = 3.675 ms; flip angle = 9˚; optimized field of view = 288 × 288 mm; slice thickness = 5 mm; and interslice gap = 0 mm.

We calculated the cross-sectional area (CSA) of the thigh in younger and older subjects from T1W and 2PD images to ensure that the same locations were being compared. As a result, CSAs of the thigh were not significantly different between T1W and 2PD values (T1W, 196.2 ± 30.7 cm$^2$; 2PD, 195.3 ± 31.0 cm$^2$). Therefore, we were able to measure the same area of the thigh using T1W and 2PD imaging (Fig 1).

### Analysis of thigh composition

MR images were read in random order for the analysis. We measured CSA of skeletal muscle and IntraMAT of the VL, adductor magnus (AM), and BF-L at the mid-thigh. Serial axial images were used to help identify muscle boundaries.

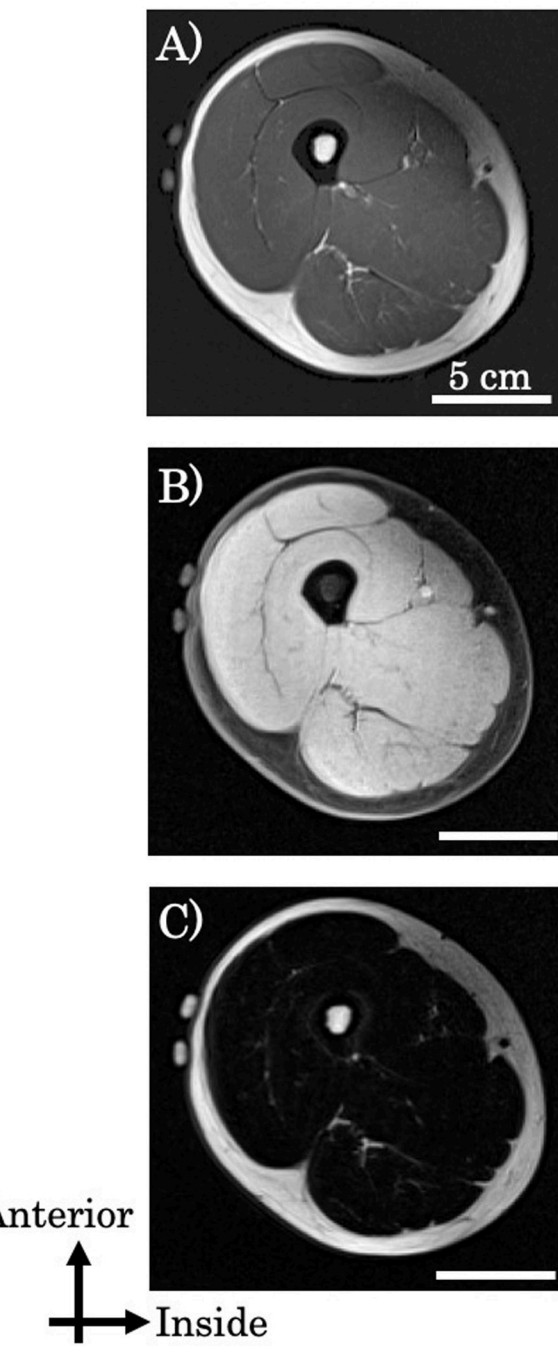

**Fig 1.** Representative T1-weighted (T1W) image (A) and 2-point Dixon (2PD) images of water (B) and fat (C).

**1) T1-weighted images.** We used the Medical Image Processing, Analysis and Visualization software (version 4.4.0; National Institutes of Health, Bethesda, MD) to analyze images on a personal computer. The MRI data analysis procedure was essentially the same as described in previously studies [13–15]. Briefly, we first corrected for image heterogeneity caused by suboptimal radiofrequency coil uniformity, or gradient-driven eddy currents, using a well-established nonparametric nonuniform intensity normalization (N3) algorithm [14, 16, 26]. This step was essential for subsequent analyses that assume homogenous signal intensities across

images. Optimized image correction parameters (N3) were determined (end tolerance, 0.0001; maximum iterations, 100; signal threshold, 1; field distance, 25 mm; subsampling factor, 4; Kernel full width, half maximum of 0.15; Wiener filter noise, 0.01), and the same parameters were applied to all images. Second, we calculated the CSA of IntraMAT content at the mid-thigh in T1W images using the threshold method, as described previously [14]. We then drew six regions of interest (ROIs) of 25 mm$^2$ each, with three ROIs on the vastus intermedius and three ROIs on the subcutaneous adipose tissue. The vastus intermedius, which is 99% skeletal muscle [12], was chosen to obtain a pure skeletal muscle peak in the pixel number–signal intensity histogram. The total number of pixels within the six ROIs was used to produce a frequency distribution and histogram of all pixels and signal intensities.

To separate muscle and adipose tissues in the pixel number–signal intensity histogram with minimal investigator bias, we implemented the Otsu threshold method, a reliable histogram-shape–based thresholding technique used in medical imaging analysis [27]. To minimize manual tracing-induced errors on thresholding values, the mean of three trials was used, and the values were applied to the VL, AM, and BF-L. After carefully tracing the edge of each muscle, the following parameters were calculated: 1) the total number of pixels within the ROI; 2) the number of pixels with a signal intensity lower than the threshold value (skeletal muscle); and 3) the number of pixels with a value higher than the threshold value (IntraMAT). The Intra-MAT content for each muscle was then calculated using the following equation:

IntraMAT content (%)
= (IntraMAT pixel numbers) / [(skeletal muscle pixel numbers) + (IntraMAT pixel numbers)] × 100

**2). Water and fat images by 2-point Dixon.** Dixon images were analyzed using ImageJ (version 1.44; National Institutes of Health, Bethesda, MD, USA). ROIs were drawn to match the corresponding T1W image voxel locations for fat and water, calculated using the 3.5 ppm chemical shift between water and lipid at 3.0 T. From these VOIs, the mean signal intensities from water and fat images by 2PD were measured to create a Dixon-based fat–water ratio using the following equation [28]:

$$\text{IntraMAT content (\%)} = 100 \times \text{Fat}_{\text{mean intensity}} / (\text{Water}_{\text{mean intensity}} + \text{Fat}_{\text{mean intensity}})$$

**Reproducibility analysis.** Manual segmentation of the thigh muscle compartments was repeated twice in 10 randomly selected subjects by one researcher with 5 years of experience in muscle image analysis to assess intra-observer reproducibility of the segmentation process. Intraclass correlation coefficient [(ICC 2.1)] in individual muscles indicated values ranging between 0.92 to 0.97 in T1W and 0.89 to 0.99 in 2PD for IntraMAT content (all $P < 0.001$).

## Statistical analysis

IntraMAT content calculated by the two MRI methods were compared using Student's unpaired $t$-test. We compared IntraMAT content by the two MRI methods between the younger and the older men using Student's unpaired $t$-test. We estimated agreement using the 95% limits of agreement method developed by Bland and Altman [29], where the difference between the adipose tissue obtained with water and fat images by T1W and 2PD imaging is plotted against their means. The limits of agreement were between 1.96 SD and -1.96 SD. All continuous variables are expressed as the mean ± SD; a two-tailed $P < 0.05$ was considered to indicate statistical significance. All statistical analyses were performed using IBM SPSS statistics (version 22.0; IBM, Tokyo, Japan).

## Results

Table 1 shows the IntraMAT content and range for the VL, AM, and BF-L. IntraMAT content in the VL determined by T1W imaging was not significantly different from that determined by 2PD; however, IntraMAT contents in the AM and BF-L were significantly higher with T1W than with 2PD imaging. In the older group, IntraMAT content was significantly higher than in the younger group in all muscles and with both the T1W and 2PD methods (Table 1).

The relationship between IntraMAT content determined using T1W and 2PD is shown in 2. There was a high positive correlation between these two values using T1W and 2PD for the different muscles (VL; r = 0.735, $P < 0.01$, AM; r = -0.717, $P < 0.01$, BF-L; r = 0.790, $P < 0.01$, VL + AM + BF-L; r = 0.686, $P < 0.01$; Fig 2).

Fig 3 shows the Bland-Altman plot of IntraMAT content for each of the three muscles determined using the T1W and 2PD methods. The limits of agreement were between 1.96 SD and -1.96 SD. However, Bland-Altman plots showed that the Pearson r (difference versus mean) was -0.698 to -0.914 ($P < 0.01$; Fig 3); therefore, a proportional bias was observed in all muscles (Fig 3). This suggests that subjects with higher IntraMAT contents showed larger differences between the two imaging modalities.

The limits of agreement in the differences between T1W and 2PD vs. the mean of the two measurements were between 1.96 SD and −1.96 SD.

Mean value, ------ 95% limits of agreement. Y = 0 is a line of perfect average agreement.

Thus, we next investigated why there were such differences in IntraMAT content between the two methods for different muscles. Fig 4 compares a representative T1W image, its respective binary image, and the fat image from 2PD for three subjects with relatively higher, middle, and lower IntraMAT contents. Comparing the arrow portions of the T1W image with the binary image and the 2PD fat image, it is clear that the portions of adipose tissue do not coincide. This tendency was observed particularly in subjects with high IntraMAT in T1W images (Figs 4 and 5). Moreover, in subjects with a large difference in IntraMAT content between T1W and 2PD, we found a difference in the thresholds in T1W and boundary values of the muscle and adipose tissue when applying the 2PD IntraMAT content to the T1W image (Fig 5).

Finally, we selected 10 subjects at random (five young subjects and five older subjects) and calculated the difference in the threshold in T1W and boundary values for muscle and adipose tissue and the values after applying the IntraMAT content of the 2PD to the T1W image for those subjects. Interestingly, there was a high negative correlation between these two values for the different muscles (VL; r = -0.874, $P < 0.01$, BF-L; r = -0.944, $P < 0.01$, VL + BF-L; r = -0.955, $P < 0.01$; Fig 6). This result indicates that the difference in IntraMAT content between T1W and 2PD methods depends on the threshold setting.

**Table 1. Effect of analysis order on intramuscular adipose tissue content (IntraMAT content; %) in T1W and 2PD images.**

| | | Mean ± SD | Younger group | Older group |
|---|---|---|---|---|
| **Vastus lateralis** | T1W | 11.0 ± 4.4 | 8.9 (4.9–13.6) | 14.1 (6.7–25.5) † |
| | 2PD | 12.2 ± 2.4 | 10.9 (8.2–13.7) | 14.0 (10.6–19.0) † |
| **Adductor magnus** | T1W | 11.6 ± 4.4 * | 8.9 (4.5–17.1) | 14.2 (9.4–24.3) † |
| | 2PD | 8.6 ± 1.7 | 7.9 (6.4–9.8) | 9.7 (7.1–14.1) † |
| **Biceps femoris-long head** | T1W | 21.3 ± 10.0 * | 16.3 (6.5–27.5) | 28.8 (5.6–45.3) † |
| | 2PD | 12.7 ± 3.2 | 11.1 (7.1–14.2) | 15.0 (9.9–21.1) † |

Abbreviations: T1W, T1-weighted; 2PD, 2-point Dixon.

* vs. 2PD, $P < 0.05$.

† vs. Younger group, $P < 0.05$.

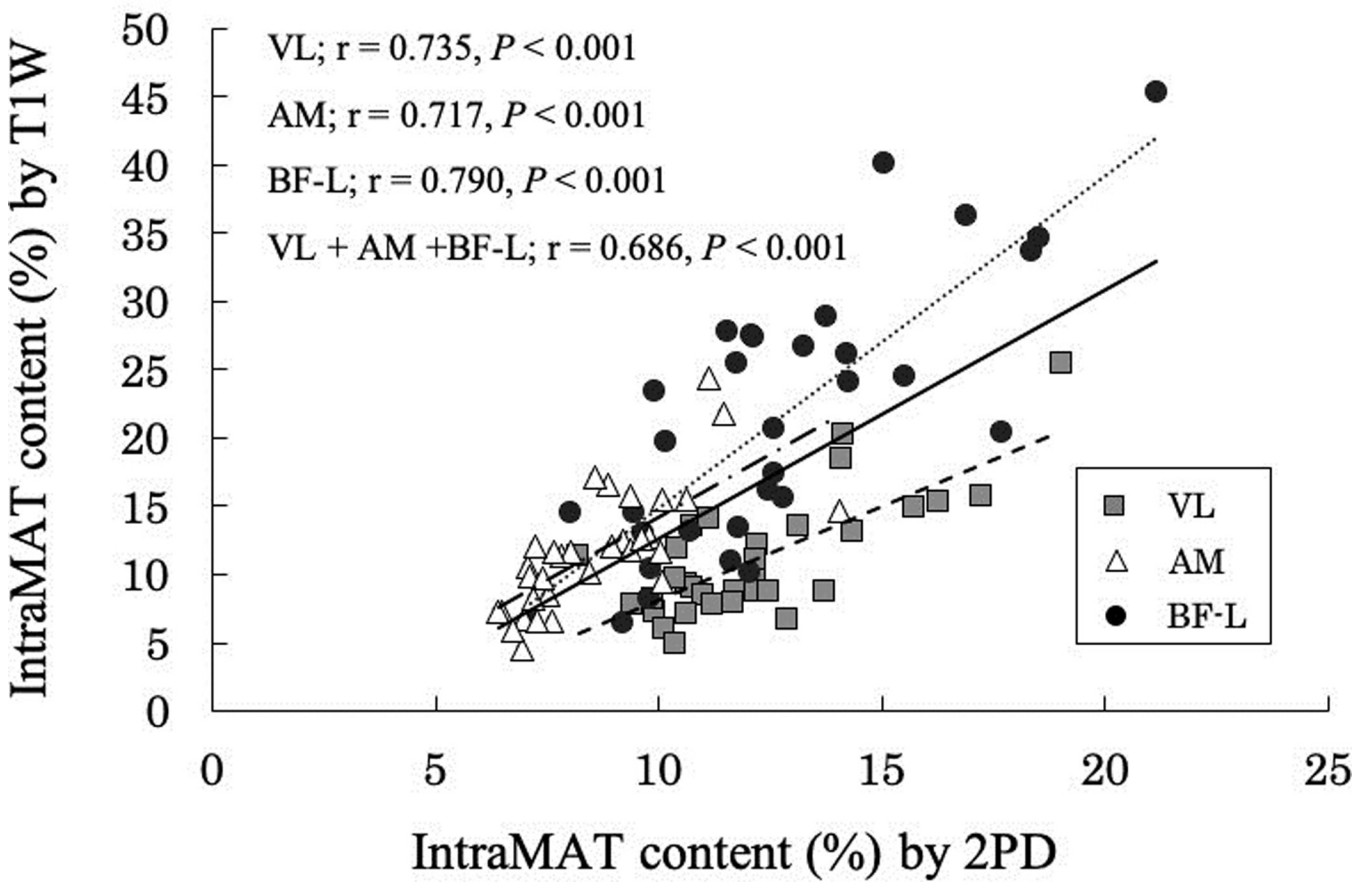

**Fig 2. Relationship between the intramuscular adipose tissue content (IntraMAT content; %) using T1-Weighted (T1W) and 2-Point Dixon (2PD) images in the Vastus Lateralis (VL), Adductor Magnus (AM), and the Biceps Femoris-Long head (BF-L).**

Difference in IntraMAT content (%) = 2PD's IntraMAT content − T1W's IntraMAT content. Difference in signal intensity (AU): Threshold of muscle and adipose tissue on T1W images − boundary value of muscle and adipose tissue when applying 2PD IntraMAT content to T1W images.

## Discussion

The main findings of this study were that: 1) IntraMAT content was significantly higher in T1W images than in 2PD images for the AM and the BF-L, but not the VL; and 2) systematic errors were found between T1W and 2PD imaging techniques that can be explained by variations in the threshold setting.

Many attempts have been made to calculate adipose tissue content within soft tissues, such as skeletal muscle, heart, and liver, using the Dixon method in healthy subjects and those with diseases [22, 30–33]. Previous studies have reported that IntraMAT content of the VL by the Dixon method ranged from 3.0 to 3.2% in younger subjects [22, 30]. However, in the present study, the IntraMAT content by 2PD for young subjects was 10.9 ± 1.2% in the VL, which was three times higher than that found in previous studies. Imaging methods (2PD or 3PD), differences in volume and cross-sectional area, subject characteristics or physical activity levels could all explain these differences in IntraMAT values.

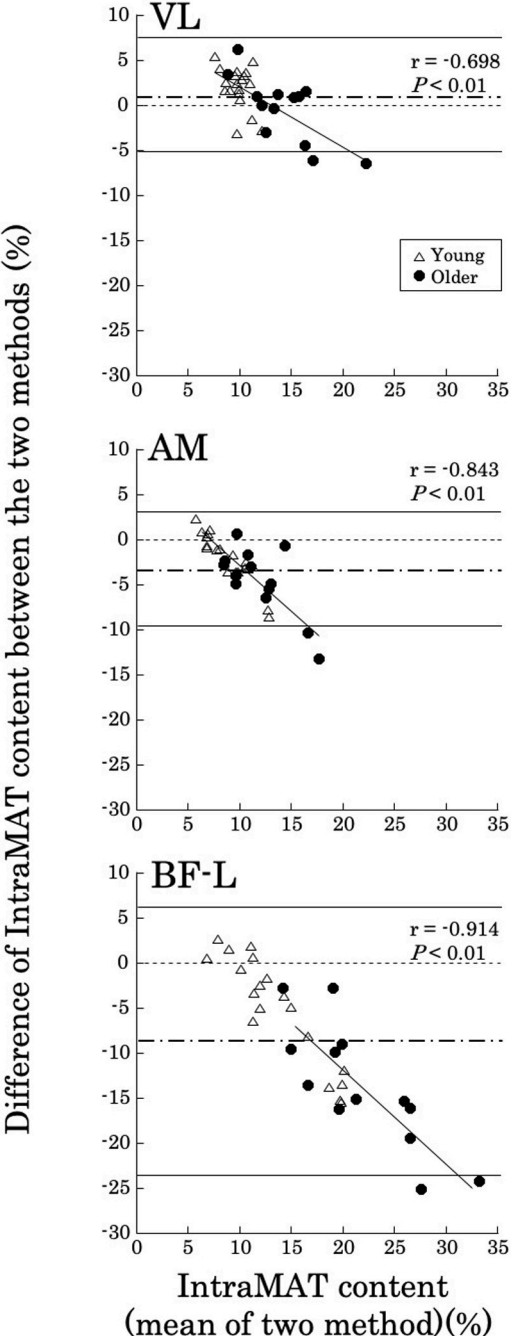

**Fig 3. Bland-Altman plots demonstrating the agreement for intramuscular adipose tissue content (IntraMAT content; %) of the Vastus Lateralis (VL), Adductor Magnus (AM) and the Biceps Femoris-Long head (BF-L) between T1-weighted and 2-Point Dixon (2PD) images.**

In the present study, the IntraMAT content by T1W was two times higher than that found in previous studies [13–15]. A previous study reported that IntraMAT contents of the VL were 3.6% and 7.7% in younger and older subjects, respectively, who were men and women [14]. Subject characteristics, physical activity levels, and sex differences could all explain these differences in IntraMAT values. We showed that IntraMAT content in the VL and BF-L were 7%

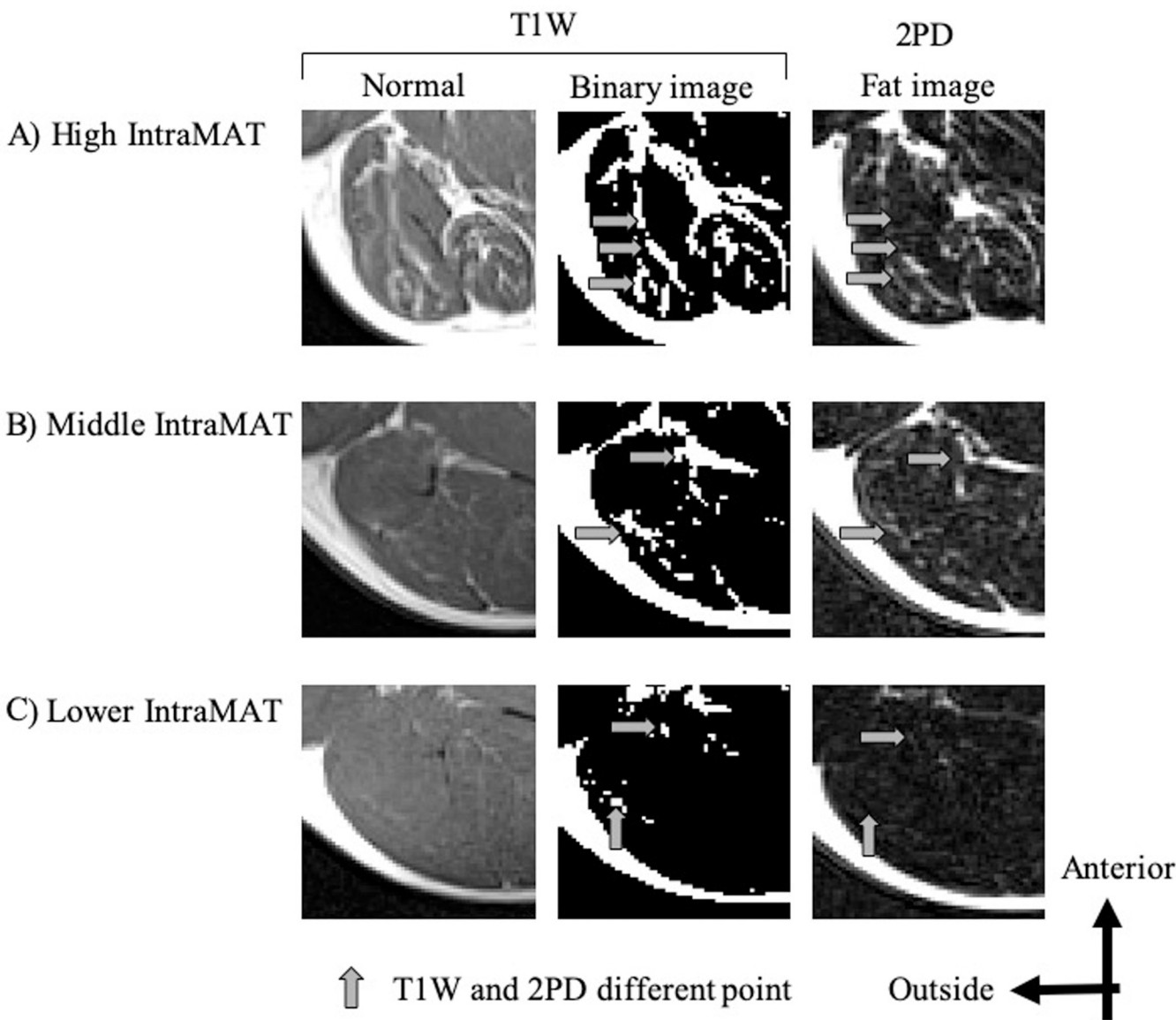

**Fig 4. Representative T1-Weighted (T1W) images and fat images by 2-Point Dixon (2PD) of the Biceps Femoris-Long head (BF-L).** Normal: T1W image after optimized image correction parameters (N3). Binary image: T1W image binarized with threshold value. Upper: subject with high IntraMAT content = 40.1% (T1W), 15.0% (2PD); Middle: subject with moderate IntraMAT content = 20.4% (T1W), 17.7% (2PD); Lower: subject with low IntraMAT content = 6.5% (T1W), 7.1% (2PD).

and 19%, respectively, in the older group, and 3% and 13%, respectively, in the younger group. The IntraMAT in the BF-L was thus 3- to 4-fold higher than that in the VL [14]. Overend and colleagues [34], who used computed tomography (CT) imaging for relatively non-contractile tissue (i.e., IntraMAT and connective tissues) within the quadriceps and hamstring muscle groups, found that IntraMAT was 3.6% and 5.4%, respectively, in younger subjects, and 7.7% and 13.6%, respectively, in older subjects. This difference in content aligns with our findings, and thus, it is a reasonable finding that the levels of adipose tissue within the VL were less than that in the BF-L. We also previously measured intramyocellular lipids (IMCL) and extramyocellular lipids (EMCL) using $^{1}$H magnetic resonance spectroscopy ($^{1}$H-MRS) in male and female subjects across a broad age range, and found collective lipid levels ranging from 9.4 to 40.2

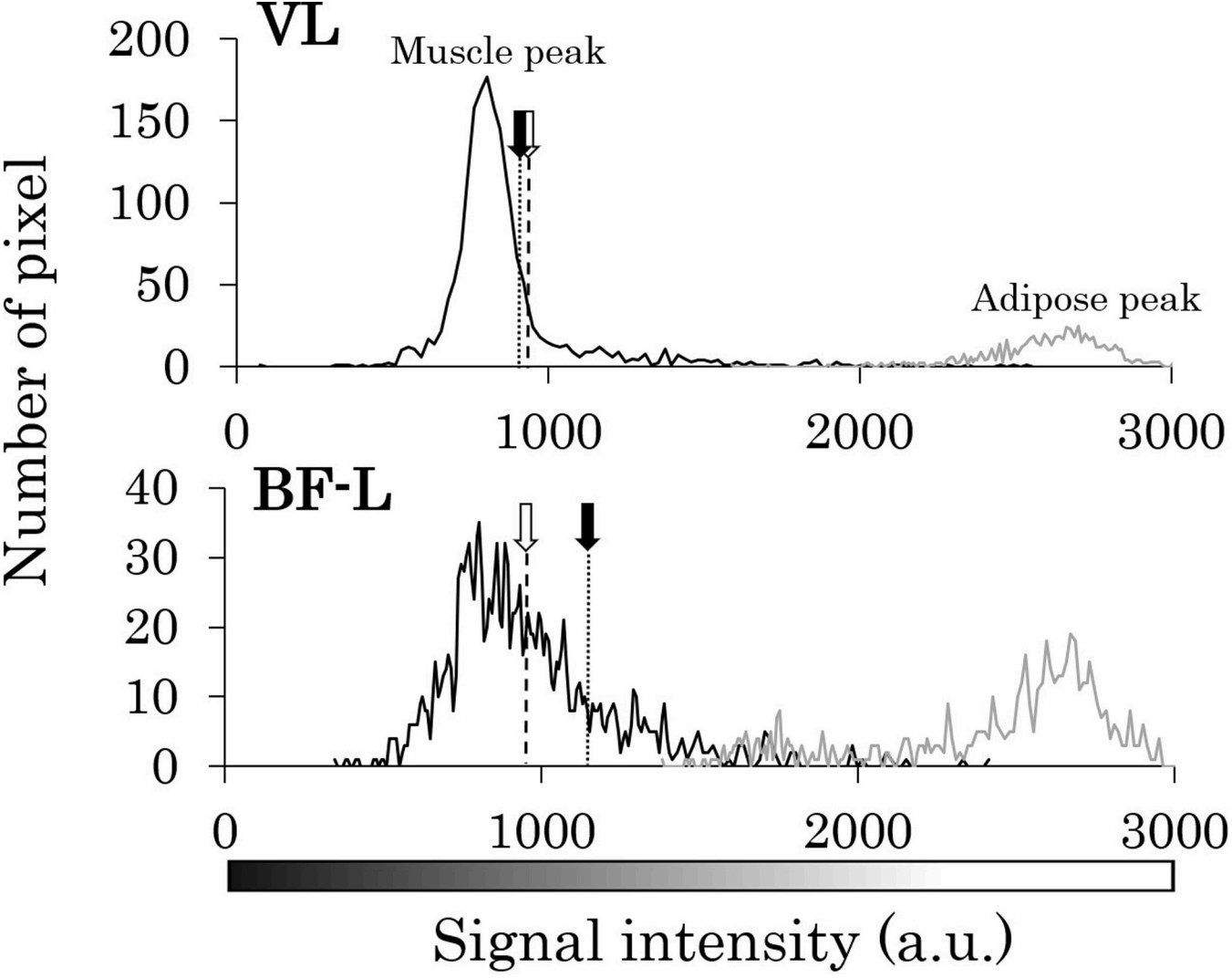

**Fig 5. Representative histograms for the Vastus Lateralis (VL) and Biceps Femoris-Long head (BF-L), created based on T1-Weighted (T1W) images.**
T1W threshold: Threshold value of muscle and adipose tissues on T1W images. 2PD boundary value: Boundary value of muscle and adipose tissues with 2PD.

mmol/kg wet weight, respectively, in the VL and from 10.5 to 94.0 mmol/kg wet weight, respectively, in the BF-L [13]. Accordingly, it is plausible that IntraMAT in the VL is likely to be lower than that in the BF-L on various imaging modalities, including MRI, CT, and $^1$H-MRS.

Although IntraMAT content in the AM and BF-L by T1W were significantly higher (by 3.0% and 8.7%, respectively) than that found by 2PD (Table 1), there was no difference in IntraMAT content for VL between the two methods, suggesting that the absolute conformity of the IntraMAT content was muscle-specific (Table 1). Thus, particular attention must be paid to the differences noted for AM and BF-L between the two methods. As shown in Fig 5, T1W and threshold-adjusted T1W images gave the same IntraMAT values as that found with 2PD; i.e., 2PD boundary value, located at the right edge of the muscle peak in the VL, indicated that almost all of the ROI was skeletal muscle using this thresholding technique. The 2PD threshold was higher than that for T1W, and this may have led to false results in terms of the amount of adipose tissue in the AM and BF-L, even though the area was categorized as skeletal muscle tissue. As a result, IntraMAT content determined by T1W was significantly higher than that determined by 2PD in the AM and BF-L.

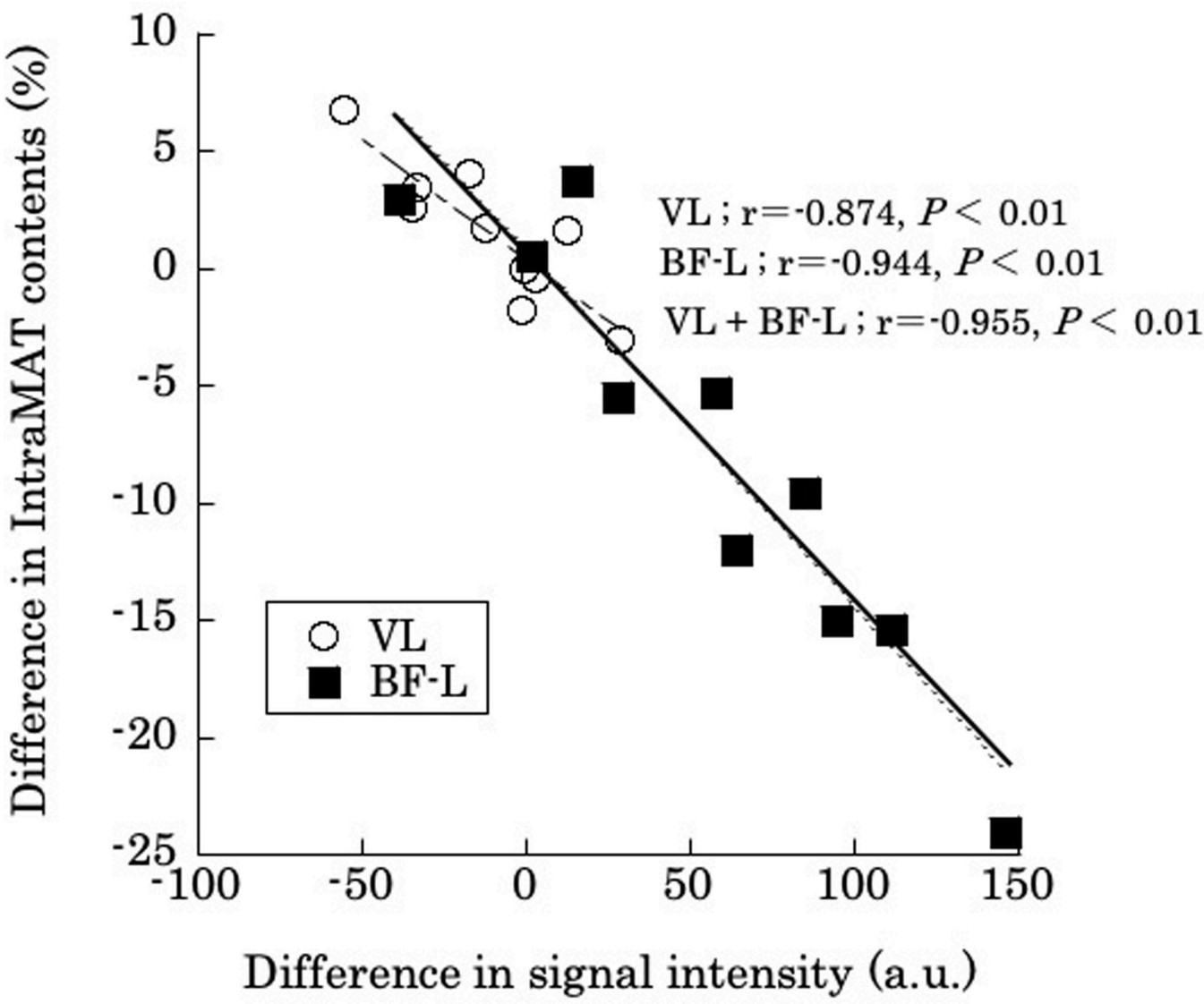

**Fig 6. Correlation for intramuscular adipose tissue content (IntraMAT content; %) between T1-Weighted (T1W) and 2-Point Dixon (2PD) images, and differences in the 2PD boundary value and T1W threshold for the Vastus Lateralis (VL) and the Biceps Femoris-Long head (BF-L).**

The further assessments conducted in Fig 6 show that the difference in IntraMAT content determined by T1W and 2PD was almost perfectly associated with differences in threshold values in randomly selected subjects; i.e. differences were dependent on signal intensity. This result suggests that the difference in IntraMAT by T1W and 2PD was closely related but dependent on threshold settings. This would be the main reason for the systematic errors in the Bland-Altman analysis of VL, BF-L and AM shown in Fig 3.

Inhomogeneity of the magnetic field in the MR system may also affect IntraMAT content values for AM and BF-L. Indeed, a previous study reported that the magnetic field on T1W images was not constant, with correction necessary to rectify for such inhomogeneity effects [14, 26]. We used the well-established N3 algorithm to correct for shading in the images due to heterogeneity linked to suboptimal radiofrequency coil uniformity or gradient-driven eddy currents [14, 26]. Although we confirm that shading was much decreased compared with the

original MR image, there is still some shading evident (e.g., from anterior to posterior or medial to lateral) [12, 14]. We expect that these persistent shading defects may have influenced our IntraMAT values. In addition, we considered that the location of muscles and imaging methods could be related changes in the signal-to-noise ratio (SNR). We measured SNR with the subtraction method, which is the most commonly used method [35]. In this method, a difference image was obtained by subtracting two identical images. We measured the SNR between T1W and water images, and T1W and fat images for 10 subjects, in the same size ROI of four corners (ROI 1 to ROI 4) to compare the SNR between the ROIs. The SNR in ROI 4 was a significantly higher than in ROI 1, but there was no significant difference between the other ROIs (Fig 7A and 7B). In this study, VL was on the ROI 3 side, AM was on the ROI 2 side, and BF-L was on the ROI 4 side. These results suggest that the effect of SNR was not observed independent of the location of muscles in this study. Unfortunately, we could not use the same voxel size between the T1W and 2PD imaging methods because of the MR device; therefore, the effect of the SNR on the results of IntraMAT could not be excluded. Furthermore, the effect of local distortion in the MR by distance of the center of the gantry to the body coil and the measurement site could also be related to the difference in IntraMAT content between T1W and 2PD. Thus, this study had some important limitations. Further human and phantom experiments under stable conditions of voxel size, SNR, and specific absorption rate would be required to reveal what caused the difference in IntraMAT content between T1W and 2PD.

Importantly, T1W has been used to measure IntraMAT in previous studies [3, 5, 8, 10, 12–16, 23–25], with a close relationship between IntraMAT content by T1W and histochemical analysis of muscle biopsies [21]. Furthermore, T1W can be used in any MRI systems from lower to higher magnetic fields. However, the Dixon method is recommended for MRI with a 1.5T or higher because of its technical limitations in being able to clearly distinguish between water and fat signals. IntraMAT content by both T1W and 2PD methods was consistent or correlated with muscle biopsy in previous studies, However, when T1W imaging is used, extra care must be taken of the IntraMAT content discrepancy with 2PD found in this study.

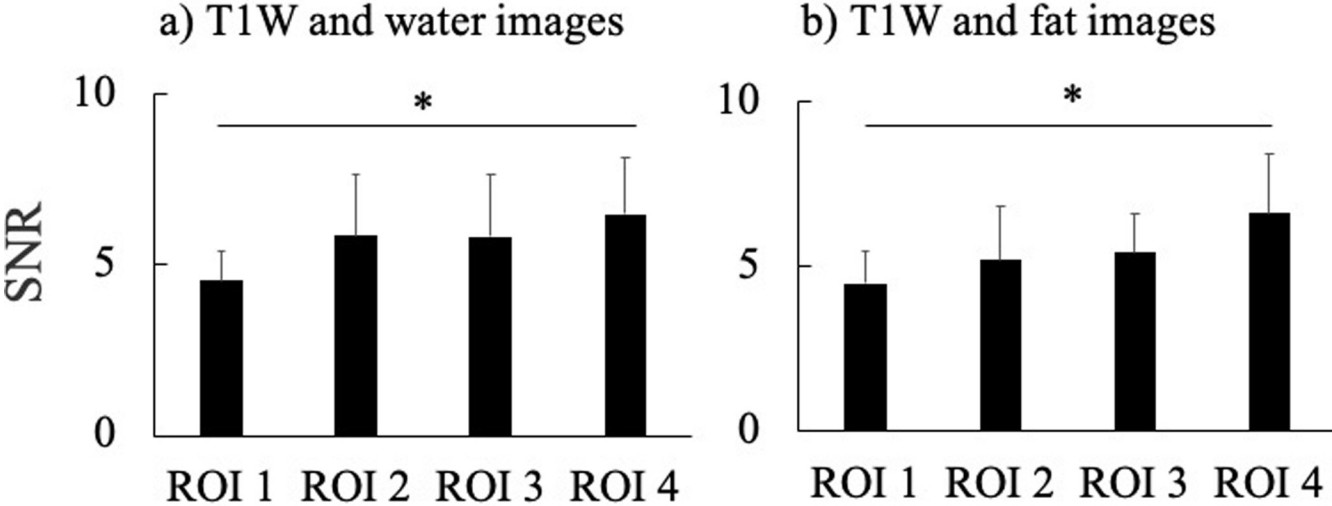

**Fig 7. Difference of four corners in the signal-to-noise ratio (SNR) between T1-Weighted (T1W) and water images, and T1W and fat images.** $^*P < 0.05$, vs. regions of interest (ROI) 1.

The discrepancy in IntraMAT contents of the T1W and 2PD methods is unknown in this study. We previously showed that IntraMAT content by T1W primarily represents EMCL, not IMCL, in the VL and BF-L muscles [13]. On one hand, this could be because a relatively 5 to 10 times larger amount of lipids accumulates in the interstitium compared with the cytoplasm of the muscle cells [13]. On the other hand, Fischer et al. (2014) reported that IntraMAT content by 2PD was closely related with the sum of IMCL and EMCL contents detected by [1]H-MRS (r = 0.918 to 0.990, $P < 0.001$) [36]. However, it has not been established that Intra-MAT contents by both T1W and 2PD methods reflected EMCL only, as shown by Akima et al. [13], or IMCL and EMCL, as shown in Fischer et al. [36] using [1]H-MRS. The major limitation in the present study was that we did not directly compare biopsy or [1]H-MRS results with MR images (T1W and 2PD), and we did not detect the exact changes in fat content (IMCL and EMCL) or total adipose tissue content. Therefore, it is important that additional research is carried out to confirm the validity of segmentation analyses for T1W and 2PD images by combining IMCL and EMCL data with [1]H-MRS or histochemical analysis.

## Conclusions

Although there was no significant difference between T1W and 2PD in IntraMAT content for the VL, the IntraMAT content using T1W was significantly higher in the AM and BF-L as compared with the 2PD method. We suggest that this is primarily because of differences in threshold settings when using T1W, particularly for measurements of the BF-L. Thus, these results suggest that care should be taken when selecting an imaging modality for IntraMAT, especially for patients who would be suspected of having higher IntraMAT values.

## Acknowledgments

The authors gratefully thank the volunteers for participation as well as Dr. Haruo Isoda, radiological technologist, Mr. Akira Ishizuka (Graduate School of Medicine, Nagoya University), and nurse, Yoko Onoda. This study was supported in part by a Grant-in-Aid for JSPS Research Fellow (to MO) and a Suzuken Memorial Foundation and Editage Grant (to NT).

## Author Contributions

**Formal analysis:** Madoka Ogawa.

**Funding acquisition:** Madoka Ogawa, Noriko Tanaka.

**Investigation:** Madoka Ogawa, Akito Yoshiko.

**Methodology:** Madoka Ogawa.

**Project administration:** Hiroshi Akima.

**Supervision:** Teruhiko Koike, Yoshiharu Oshida, Hiroshi Akima.

**Writing – original draft:** Madoka Ogawa.

**Writing – review & editing:** Akito Yoshiko, Noriko Tanaka.

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
