## [Decision Letter · Decision Letter 0]

14 Nov 2019

PONE-D-19-29508

Comparing intramuscular adipose tissue on T1-weighted and two-point Dixon images

PLOS ONE

Dear Dr. Ogawa,

Thank you for submitting your manuscript to PLOS ONE. After careful consideration, we feel that it has merit but does not fully meet PLOS ONE’s publication criteria as it currently stands. Therefore, we invite you to submit a revised version of the manuscript that addresses the points raised during the review process.

We would appreciate receiving your revised manuscript by Dec 29 2019 11:59PM. To enhance the reproducibility of your results, we recommend that if applicable you deposit your laboratory protocols in protocols.io, where a protocol can be assigned its own identifier (DOI) such that it can be cited independently in the future. For instructions see: http://journals.plos.org/plosone/s/submission-guidelines#loc-laboratory-protocols

We look forward to receiving your revised manuscript.

Kind regards,

Kiyoshi Sanada, PhD

Academic Editor

PLOS ONE

Journal Requirements:

2. Please provide additional details regarding participant consent. In the ethics statement in the Methods and online submission information, please ensure that you have specified who gave consent and whether it was informed.

Reviewers' comments:

Reviewer's Responses to Questions

**Comments to the Author**

1. Is the manuscript technically sound, and do the data support the conclusions?

Reviewer #1: Yes

Reviewer #2: Yes

2. Has the statistical analysis been performed appropriately and rigorously? 

Reviewer #1: Yes

Reviewer #2: Yes

3. Have the authors made all data underlying the findings in their manuscript fully available?

Reviewer #1: Yes

Reviewer #2: Yes

4. Is the manuscript presented in an intelligible fashion and written in standard English?

Reviewer #1: Yes

Reviewer #2: Yes

5. Review Comments to the Author

Reviewer #1: In this manuscript, the author aimed to compare intramuscular adipose tissue content determined by two-point Dixon imaging and T1-weighted imaging, calculated using thresholding techniques. This study showed that there was no significant difference between T1W and 2PD in IntraMAT content for the VL, whereas the IntraMAT content using T1W was significantly higher in the AM and BF-L as compared with the 2PD method. Additionally, this is primarily because of differences in threshold settings when using T1W. This manuscript may be potentially interesting and clinical significance, however, there are several key concerns that need to be addressed.

Major comments:

1. The authors concluded that T1W imaging is likely to be the better option for IntraMAT measurements. It is ambiguous about the process of this conclusion. Therefore, the authors need to discuss the more clearly. IntraMAT content determined by T1W was significantly higher than that determined by 2PD in the AM and BF-L. Which value is appropriate?

2. What is the gold standard to assess IntraMAT? T1W? Dixon? biopsies? Please describe in the introduction.

3. Is the order of measurement randomized? Can the authors clarify the study protocols?

4. In the Table 1, the authors should present the comparison between young and older. It will be interesting to know whether the effects of aging differ between T1W and 2PD methods.

5. Instead of showing only the relationship between difference in IntraMAT contents and difference in signal intensity, Figure should present the relationship between IntraMAT content determined by T1W and 2PD

6. In the Figure 3, is there the representative image of Lower IntraMAT?

7. the authors described “the IntraMAT content of this study by 2PD which was three times higher than that found in previous studies”. How is T1W compared to previous studies? Please discuss.

Minor comments:

1. Please change “positive correlation” to “negative correlation” (page 14, line 7).

Reviewer #2: The paper brings some original data and I have read it with a great interest. Nevertheless I have few comments and suggestions:

The MR image shows a change in contrast when there is a distance change from the magnetic field center in the gantry.

Therefore, in addition to this experimental data, 1) Change the subject's imaging position, 2) After imaging at a site away from the center of the bore, recalculate to see if similar results are obtained.

In addition, the calculated MR signal intensity depends on the voxel size. Therefore, change the Voxel size and verify that similar results are obtained.

6. PLOS authors have the option to publish the peer review history of their article (what does this mean?). If published, this will include your full peer review and any attached files.

Reviewer #1: No

Reviewer #2: No

---

## [Author Response · Author response to Decision Letter 0]

5 Jan 2020

Journal Requirements:

Q1. When submitting your revision, we need you to address these additional requirements.

A1. I confirmed PLOS ONE style templates and changed our maniscript.

Q2. Please provide additional details regarding participant consent. In the ethics statement in the Methods and online submission information, please ensure that you have specified who gave consent and whether it was informed.

A2.I added Ethics Statement in online and manuscript. However, if you need to more information, please tell me.

Q3. We note that you have indicated that data from this study are available upon request. PLOS only allows data to be available upon request if there are legal or ethical restrictions on sharing data publicly. For information on unacceptable data access restrictions, please see http://journals.plos.org/plosone/s/data-availability#loc-unacceptable-data-access-restrictions.

A3.Our data contain potentially identifying information and ethics committee didn't allow the connecting internet those data. Data are available from Ethics Committee for researchers who meet the criteria for access to confidential data.

Ethics committee of Nagoya University Graduate School of Medicine

Address: Nagoya University 65 Tsurumai, Showa-ku, Nagoya, Aichi 466-8550, Japan

E-mail: ethics@med.nagoya-u.ac.jp

Q4. Please amend either the abstract on the online submission form (via Edit Submission) or the abstract in the manuscript so that they are identical.

A4. We apologize for this oversight. We confirmed and changed our abstract in manuscript and online.

Q5. Your ethics statement must appear in the Methods section of your manuscript. If your ethics statement is written in any section besides the Methods, please move it to the Methods section and delete it from any other section. Please also ensure that your ethics statement is included in your manuscript, as the ethics section of your online submission will not be published alongside your manuscript.

A5. We deleted ethics statement in end of our manuscript. We written ethics statement in Methods.

---

## [Decision Letter · Decision Letter 1]

29 Jan 2020

PONE-D-19-29508R1

Comparing intramuscular adipose tissue on T1-weighted and two-point Dixon images

PLOS ONE

Dear Dr. Ogawa,

Thank you for submitting your manuscript to PLOS ONE. After careful consideration, we feel that it has merit but does not fully meet PLOS ONE’s publication criteria as it currently stands. Therefore, we invite you to submit a revised version of the manuscript that addresses the points raised during the review process.

We would appreciate receiving your revised manuscript by Mar 14 2020 11:59PM. To enhance the reproducibility of your results, we recommend that if applicable you deposit your laboratory protocols in protocols.io, where a protocol can be assigned its own identifier (DOI) such that it can be cited independently in the future. For instructions see: http://journals.plos.org/plosone/s/submission-guidelines#loc-laboratory-protocols

We look forward to receiving your revised manuscript.

Kind regards,

Kiyoshi Sanada, PhD

Academic Editor

PLOS ONE

Reviewers' comments:

Reviewer's Responses to Questions

**Comments to the Author**

1. If the authors have adequately addressed your comments raised in a previous round of review and you feel that this manuscript is now acceptable for publication, you may indicate that here to bypass the “Comments to the Author” section, enter your conflict of interest statement in the “Confidential to Editor” section, and submit your "Accept" recommendation.

Reviewer #1: All comments have been addressed

Reviewer #2: All comments have been addressed

2. Is the manuscript technically sound, and do the data support the conclusions?

Reviewer #1: Yes

Reviewer #2: Yes

3. Has the statistical analysis been performed appropriately and rigorously? 

Reviewer #1: Yes

Reviewer #2: Yes

4. Have the authors made all data underlying the findings in their manuscript fully available?

Reviewer #1: Yes

Reviewer #2: Yes

5. Is the manuscript presented in an intelligible fashion and written in standard English?

Reviewer #1: Yes

Reviewer #2: Yes

6. Review Comments to the Author

Reviewer #1: The authors have addressed my major concerns, and the manuscript has much improved. I have no further comments for the authors.

Reviewer #2: I understand your response.

It can be inferred from the text that T1w and the 2PD method was performed continuously. Since it is described as "consective," it is not necessary to describe it as "continuously."

As you know and use it, MR devices have different image data and contrast (power from changes in SN and SAR) depending on continuous imaging and shimming conditions. These problems have been corrected in the image conversion process. If it is far from the center of the gantry, the effect of local distortion, due to MR, also appears.

It is not a matter of continuity, but the MR system. In addition, the distance between the body-coil and the measurement site is also an issue.

This paper is a valuable paper that utilizes an uncomplicated evaluation and hopes for future development. Therefore, it is essential that the reliability of the published data is established and whether the reader can obtain comparative data by the same method. I understand your response is giving up its verification.

The previous advice does not consider continuous imaging as a problem but points out the reproducibility of data. I highly recommend you to describe problems with MR equipment properly. The advice for voxel size is the same.

These points should also be considered in the limitations.

If you use a method that can be evaluated easily, it is recommended that you consider it properly.

7. PLOS authors have the option to publish the peer review history of their article (what does this mean?). If published, this will include your full peer review and any attached files.

Reviewer #1: No

Reviewer #2: No

---

## [Author Response · Author response to Decision Letter 1]

11 Mar 2020

Dear Dr. Sanada and Reviewer #2 

We wish to express our deep appreciation to the reviewer for the insightful comment on our manuscript. We apologize for not answering your previous question perfectly. 

We have worked hard to incorporate your feedback and hope you will agree that that these revisions have made our manuscript suitable for publication in PLOS ONE.

Sincerely,

Madoka

---

## [Decision Letter · Decision Letter 2]

18 Mar 2020

Comparing intramuscular adipose tissue on T1-weighted and two-point Dixon images

PONE-D-19-29508R2

Dear Dr. Ogawa,

We are pleased to inform you that your manuscript has been judged scientifically suitable for publication and will be formally accepted for publication once it complies with all outstanding technical requirements.

With kind regards,

Kiyoshi Sanada, PhD

Academic Editor

PLOS ONE

Additional Editor Comments (optional):

Reviewers' comments:

Reviewer's Responses to Questions

**Comments to the Author**

1. If the authors have adequately addressed your comments raised in a previous round of review and you feel that this manuscript is now acceptable for publication, you may indicate that here to bypass the “Comments to the Author” section, enter your conflict of interest statement in the “Confidential to Editor” section, and submit your "Accept" recommendation.

Reviewer #1: All comments have been addressed

Reviewer #2: All comments have been addressed

2. Is the manuscript technically sound, and do the data support the conclusions?

Reviewer #1: Yes

Reviewer #2: Yes

3. Has the statistical analysis been performed appropriately and rigorously? 

Reviewer #1: Yes

Reviewer #2: Yes

4. Have the authors made all data underlying the findings in their manuscript fully available?

Reviewer #1: Yes

Reviewer #2: Yes

5. Is the manuscript presented in an intelligible fashion and written in standard English?

Reviewer #1: Yes

Reviewer #2: Yes

6. Review Comments to the Author

Reviewer #1: (No Response)

Reviewer #2: I have no further comments for the authors.

I read the revised version. Although the experimental data can be easily measured by using the MR device, the verification of the data becomes uncertain if the experiment is performed without understanding its essence. I think the paper is more polite than the previous one.

I am looking forward to reading your future paper.

7. PLOS authors have the option to publish the peer review history of their article (what does this mean?). If published, this will include your full peer review and any attached files.

Reviewer #1: No

Reviewer #2: No

---

## [Editor Report · Acceptance letter]

23 Mar 2020

PONE-D-19-29508R2 

Comparing intramuscular adipose tissue on T1-weighted and two-point Dixon images 

Dear Dr. Ogawa:

I am pleased to inform you that your manuscript has been deemed suitable for publication in PLOS ONE. Congratulations! Your manuscript is now with our production department. 

With kind regards,

on behalf of

Dr. Kiyoshi Sanada 

Academic Editor

PLOS ONE